# Emerging Postharvest Technologies to Enhance the Shelf-Life of Fruit and Vegetables: An Overview

**DOI:** 10.3390/foods11233925

**Published:** 2022-12-05

**Authors:** Michela Palumbo, Giovanni Attolico, Vittorio Capozzi, Rosaria Cozzolino, Antonia Corvino, Maria Lucia Valeria de Chiara, Bernardo Pace, Sergio Pelosi, Ilde Ricci, Roberto Romaniello, Maria Cefola

**Affiliations:** 1Department of Science of Agriculture, Food and Environment, University of Foggia, Via Napoli, 25, 71122 Foggia, Italy; 2Institute of Sciences of Food Production, National Research Council of Italy (CNR), c/o CS-DAT, Via Michele Protano, 71121 Foggia, Italy; 3Institute on Intelligent Industrial Systems and Technologies for Advanced Manufacturing, National Research Council of Italy (CNR), Via G. Amendola, 122/O, 70126 Bari, Italy; 4Institute of Food Science, National Research Council (CNR), Via Roma 64, 83100 Avellino, Italy

**Keywords:** active packaging, cold plasma, dipping, E-nose, high hydrostatic pressure, image analysis, innovative postharvest technologies, pulsed electric field, vacuum impregnation, near-infrared spectroscopy

## Abstract

Quality losses in fresh produce throughout the postharvest phase are often due to the inappropriate use of preservation technologies. In the last few decades, besides the traditional approaches, advanced postharvest physical and chemical treatments (active packaging, dipping, vacuum impregnation, conventional heating, pulsed electric field, high hydrostatic pressure, and cold plasma) and biocontrol techniques have been implemented to preserve the nutritional value and safety of fresh produce. The application of these methodologies after harvesting is useful when addressing quality loss due to the long duration when transporting products to distant markets. Among the emerging technologies and contactless and non-destructive techniques for quality monitoring (image analysis, electronic noses, and near-infrared spectroscopy) present numerous advantages over the traditional, destructive methods. The present review paper has grouped original studies within the topic of advanced postharvest technologies, to preserve quality and reduce losses and waste in fresh produce. Moreover, the effectiveness and advantages of some contactless and non-destructive methodologies for monitoring the quality of fruit and vegetables will also be discussed and compared to the traditional methods.

## 1. Introduction

Fresh fruit and vegetables are important sources of essential vitamins, minerals, and antioxidants. In recent decades, consumers have become more likely to consume these products since they are aware of their potential preventive effects against some non-communicable diseases. In addition, increasing safety (chemical, toxicological and microbial) and traceability are important aspects for all the players in the supply chain (from the farm to consumers) [1]. Food quality may be defined as the combination of several physical and chemical attributes (appearance, texture, flavor, and nutritional value) which have a crucial impact on determining the degree of consumer acceptability [2]. The quality of fruit and vegetables is mostly based on the evaluation of different external attributes (size, shape, color, gloss, firmness, texture, and taste) and internal factors (chemical, physical and microbial) dealing with nutritional traits, safety, and sustainability [3].

Harvested commodities are metabolically active and highly perishable. They undergo quality losses due to the ripening and senescence processes, which are often associated with the development of spoilage microorganisms and other undesired phenomena, which must be controlled to preserve the quality and increase the shelf-life of the product during storage [4,5]. Furthermore, high water activity and the presence of nutritional factors associated with these matrices can also favor the growth of pathogens [6,7]. Fruit and vegetables with better sensory and nutritional attributes have a relevant economic value. Consequently, inadequate preservation practices, besides causing important losses in nutritional and quality characteristics, can have a detrimental economic impact along the entire supply chain, from growers to consumers. The Food and Agriculture Organization (FAO) estimated that 33% of the total food produced for human consumption is lost due to postharvest spoilage (Figure 1). Overall, 44% of losses occur in industrialized (developed) countries and 40% occur in developing countries [8].

In a recent report released by the FAO [10], it was reported that fruit and vegetables are the food group with the second-highest value of losses and waste (about 22%), exceeded only by roots, tubers, and oil-bearing crops at all stages in the food supply chain (Figure 2), due to their highly perishable nature.

The reduction of losses and wastage in fruit and vegetable production is important for improving security, promoting the sustainability of the environment, and reducing production costs.

Optimal postharvest handling, including storage time and temperature management, relative humidity, chemical and/or physical treatments, and packaging (i.e., a modified atmosphere) can slow down the biological processes caused by senescence and maturation, reduce or inhibit the development of physiological disorders, and minimize microbial growth and contamination [1,11,12]. For example, temperature is the most important and simplest postharvest factor that can delay the decay of the product [2,13]. Generally, there is a recommended storage temperature for each product, and optimum quality preservation can be achieved if the commodity is rapidly cooled to its best postharvest storage temperature as soon as possible after harvest [14].

Furthermore, the amounts of internationally traded agricultural products have generally increased with the progress of food globalization. Consequently, the distance and the duration of transport have been extended; therefore, prolonged exposure to non-optimal storage conditions can cause a rapid deterioration in the product quality, with the consequent losses and waste. For these reasons, increased product shelf-life and more advanced quality control technology over longer distances have become key issues [15]. 

Nowadays, there is much research about the effects of new postharvest physical and chemical treatments and about the biocontrol techniques used to preserve the quality, nutritional value, and safety of fresh produce, from harvest to consumer consumption. It has been widely demonstrated that these methods, whether alone or in combination with the appropriate management of storage temperature, could preserve the principal quality and nutritional traits of fruit and vegetables. 

Conventional methods (sensory evaluations and analytical methodologies) used to evaluate fruit and vegetable quality are destructive, time-consuming, and labor- and cost-intensive. Moreover, these techniques are not suitable for in-line application in industrial or in market settings to give real-time information to the consumers on the quality of the product at hand. On the contrary, the emerging contactless and non-destructive technologies for the quality monitoring of fresh produce, including near-infrared spectroscopy, hyperspectral or multispectral imaging, image analysis, electronic noses, etc., present numerous advantages over conventional destructive methods [3]. These emerging techniques are normally used for external and internal quality evaluation and are principally based on the measurement of chemical or physical attributes that correlate with certain quality traits of agricultural products [16].

The aim of this review article is to report a detailed collection of the research works carried out in the last years about the principal application of new postharvest physical and chemical emerging technologies to preserve the quality and reduce postharvest losses and waste in fresh fruit and vegetables. Moreover, the effectiveness and advantages of quality evaluation via some contactless and non-destructive methodologies are described and compared to conventional approaches.

## 2. Postharvest Strategies to Extend the Shelf-Life of Fruit and Vegetables

### 2.1. Physical Treatments

Emerging non-thermal physical technologies have moved into the spotlight in the last few years, with the aim of replacing the traditional postharvest technologies based on thermal processing. Besides being highly water-consuming, the conventional methodologies can show deleterious effects on the fresh commodities’ quality aspects. These novel technologies can reduce nutrient losses, increase consumer acceptability, promote food quality, and prolong shelf-life and freshness, guaranteeing the complete absence of chemical byproducts in the processed product, combined with a reduction in the environmental impact [17]. Among these emerging methods, microwave heating, high hydrostatic pressure, pulsed electric fields, high hydrostatic pressure, and cold plasma have been applied to reduce the microbial load, thus helping to preserve the freshness and quality characteristics of fruit and vegetables [18]. However, these techniques show numerous advantages and disadvantages that have begun to be investigated to achieve a suitable quality standard by adopting cost-effective methods [19]. 

This subsection aims to briefly collect the emerging physical technologies that have been applied to fresh and minimally processed or fresh horticultural products over the past few years, particularly focusing on the treatments’ effectiveness in maintaining the quality and safety of fruit and vegetables.

#### 2.1.1. Microwave

Generally, heating processes, such as hot water and hot air treatments, high-temperature/short-time treatments, and radio frequency can cause a reduction in the contents of essential nutrients and flavor-related compounds, due to heat application and its slow distribution in plant tissue by means of conduction or conduction processes. Microwaving was applied in this context as an alternative to conventional heating [20], with the aim of achieving a fast and effective increase in temperature without a temperature gradient. This technique briefly treats fruits and vegetables to manage microbial growth throughout the product’s minimal processing, minimizing losses of quality and simultaneously guaranteeing the smallest effect on the environment and the absence of residues in the treated product [21].

However, little information is available in the literature related to the application of this innovative physical technique with the aim of minimizing quality loss during the postharvest phase (Table 1). Minimally processed carrots [22], apples, and bok choy (*Brassica campestris* L.) were subjected to high-power/short-time treatments, demonstrating promising results from a microbiological point of view [23,24]. An excessive duration or intensity of the microwave treatment, however, can induce excessive temperature increase, damaging the fresh tissue, due to non-uniform heating [25]. For this reason, there is a need to overcome several challenges to achieve a successful microwave process on an industrial scale for fresh and minimally processed products.

#### 2.1.2. Pulsed Electric Field

Recently, pulsed electric field (PEF) technology has become of the most interest because of its capability to obtain safe food with minimal heat production through the use of μs to ms-pulses of a high electric field of high intensity [50]. This technique has been widely used on liquid, semi-solid and solid foods, also including fresh fruit, vegetable smoothies, and juices. The PEF parameters to be optimized to obtain microbial and enzymatic inactivation in fresh products are represented by the strength of the electric fields, the treatment time, and the frequency, polarity, or shape of the pulses. Beneficial effects from PEF treatment were observed in the reduction of enzymatic activity with a consequent improvement in the quality parameters [27,28,29,51,52,53], as reported in Table 1. PEF-treated products better retained their fresh flavor, textural, and functional attributes, including a longer shelf-life and greater microbiological safety. However, PEF-induced metabolic stress could negatively affect the quality of the final product, thus limiting the application of PEF to fresh-cut products. Among the limited studies reporting the effects on the metabolism and characteristics of minimally processed produce, Li et al. [26] have reported that PEF treatment limited the browning index and acrylamide content in ready-to-eat lotus root during postharvest life.

#### 2.1.3. High Hydrostatic Pressure

High hydrostatic pressure (HHP) technology is mostly used for microbial inactivation or reduction and for enzyme denaturation. However, high pressure, inducing injuries on microbial cellular structures, might show similar effects on the plant cells; thus, an in-depth study into treatment optimization in various fresh systems is required. Several results show that HHP significantly affects microbial load; however, it also influences the functionality of proteins, such as enzymes and tissue structure, specifically and differentially due to the wide variety of product types [30]. Moreover, the stimulation and accumulation of nutraceutical compounds were also observed. The effects of HHP application have been reported for different minimally processed horticultural commodities [31,32,34,37], whole produce [30,35,36,46], and juice [54], demonstrating great efficiency in improving food safety aspects and in maintaining quality.

#### 2.1.4. Cold Plasma

The application of cold plasma is widely used in the whole and minimally processed fruit and vegetable industry as an innovative technology used to handle microbial development [55], with the aim of replacing conventional sanitation treatments, meanwhile preserving the nutritional and antioxidant aspects of food products. Several scientific studies reported the effectiveness of non-thermal plasma on different horticultural products (Table 1). Several fruit-based fresh-cut products have been subjected to plasma treatment with beneficial effects, in terms of the quality parameters and the inhibition of microbial growth. During the last few years, plasma-activated water (PAW) application has also been more and more widely studied. This technique allows the producer to avoid cell damage caused by direct exposure to cold plasma, representing a valuable alternative to the conventional solution of washing during fresh-cut processing for several products. To date, cold plasma and PAW washing applications have been described for strawberries [41,42,56,57,58], kumquat fruit [59,60] green leafy vegetables [45,57,61,62,63,64], blueberries [46,65,66,67], fresh-cut apples [38,39,48,49,68,69] pears [44,70], cantaloupe melons [40], mushrooms [47,71], tomatoes [72], kiwifruits [73], and red currants [74].

### 2.2. Dipping and Vacuum Impregnation

The quality of fresh-cut fruit and vegetables is widely affected by physiological changes, such as enzymatic browning caused by tissue damage and high respiration rates, and by physical factors, including mechanical injuries and the removal of outer protective coverings, which culminate in faster weight loss, shriveling, loss of color and appearance, and shorter shelf-life [75]. Hence, innovative food processing technologies, such as dipping and vacuum impregnation techniques, are being investigated and implemented to sanitize, reduce enzymatic browning, improve the texture, and use nutrients (vitamins, probiotics, minerals, organic acids, phenols, etc.) to fortify fresh-cut fruit and vegetables, in order to preserve and improve the quality and to extend the shelf-life of these products [76,77,78]. 

Dipping treatments consist of soaking the product, with or without mechanical agitation, followed by the removal of the excess solution. This method is commonly used on whole, peeled, shredded, and sliced commodities and on more perishable products, as it favors the dispersion of the solution, covering the maximum surface area of the product without any damage or stress [79]. One of the major advantages of these dipping treatments is the removal of the cellular exudates, which can have a detrimental effect on the postharvest quality of commodities. Depending on the food product, the variables of the dipping process to be optimized are the time of soaking, the frequency, the solute composition, the temperature, and the concentration of the solution. Several studies have tested dipping treatments with calcium (Ca) salts to extend the shelf-life of products. Ca enrichment, in fact, has several advantages, including the reduction of microbial growth due to a decrease in activity water, the improvement of texture, acceptability, and storability, and the prevention of browning due to the oxidation phenomena and the development of off-flavors in fresh-cut foods [80,81,82,83]. Giacalone and Chiabrando [84] compared the effect of dipping in Ca salts (chloride or propionate) and citric acid solutions on apple cubes, highlighting an improvement in firmness, color, sensory attributes, and browning inhibition on fruits treated with Ca chloride and citric acid. Albertini et al. [85] tested the immersion of papaya slices in a water solution containing cinnamaldehyde, Ca chloride, and their combinations, showing that a combination treatment increased the maintenance of firmness and had an effect on flavor and taste. Other studies have evaluated the dipping process in solutions of Ca chloride, combined with pectin methyl-esterase (PME), on firmness and some of the quality attributes of raspberry fruits during refrigerated storage, showing that the treatment improved the firmness and reduced the weight loss of the fruit [86]. The application of natural extracts for use as anti-browning agents (polyphenols, carotenoids, organic acids, and bioactive peptides) represents a recent approach that is used to improve the quality and extend the shelf-life of fresh-cut fruit and vegetables [87]. Many natural agents from tomato skin [88], pineapple juice [89], pomegranate peel [90], mango peel [91], aloe vera gel [92], and extracts from pumpkin, artichoke, grape, and broccoli, etc. [93] have been reported for browning control in fresh-cut fruits. Supapvanich et al. [94] tested the immersion of fresh-cut apple slices in coconut water and evaluated several quality attributes (visual quality, color, and enzymatic and antioxidant activity) during storage, reporting that coconut-liquid endosperms could inhibit the browning incidence of fresh-cut fruit for up to 9 days at 4 °C. Wessels et al. [93] evaluated the anti-browning effect of 36 plant extracts, applied as dipping solutions on the enzymatic activity and the color parameters of fresh-cut apple slices, suggesting that the inhibition of browning might be attributable to the antioxidant activity of secondary plant metabolites, especially the phenolic compounds.

Food vacuum impregnation (VI) (Figure 3) is a method that allows producers to directly introduce, dissolve, or suspend substances in the void fraction (i.e., the pores) of a food matrix in a controlled manner [95]. VI includes two main steps: (1) the pressure is reduced in the system (under vacuum), the native gases and liquids are removed, and the product pores are expanded under the action of pressure gradients until mechanical equilibrium is achieved; (2) the atmospheric pressure is restored (relaxation period) and, with the opposite pressure gradient, the external solution fills the pores while the tissues relax, until a new equilibrium has been reached. The hydrodynamic mechanism and deformation-relaxation phenomena take place during the vacuum impregnation process, leading to the flow of external solutions into the intracellular spaces of foods [96].

Before the application of VI treatment, it is necessary to consider the porosity, the tissue structure, the size and geometry of the food, the impregnation solution (concentration and type of solute), and the process parameters (vacuum pressure, exposure time, relaxation time at atmospheric pressure, temperature, product/solution relationship, and agitation). Fruit and vegetables have a substantial amount of intercellular space occupied by gas; therefore, the VI represents a suitable tool for incorporating compounds that allow producers to extend the shelf-life without modifying the cellular structure of food [97]. In the last few years, several authors have studied the application of VI to obtain foods enriched with antimicrobial, functional, and structural compounds. Kang and Kang [98] evaluated the effect of VI when applied to the washing process, with a malic acid solution used to remove microorganisms from the surface of broccoli. Yılmaz and Ersus Bilek [99] evaluated the simultaneous effect of an ultrasound-VI process, using natural phenolic compounds extracted from black carrot, on several quality attributes and on the inhibition of microorganism growth in fresh-cut apple discs. Santana Moreira et al. [100] enriched minimally processed fruit salad with β-carotene and lutein, which are natural pigments used to improve the appearance and color of food products, as well as to offer health benefits related to antioxidant and anticancer activity. Natural fruit and vegetable juices, rich in bioactive compounds, were used to impregnate solid fruits and generate products with high nutritional value [101,102]. Derossi et al. [103], using the VI technology, enriched fresh-cut apple slices to improve their antioxidant activity and healthy properties by filling their pores with an aloe vera gel extract; other authors used grape juice to increase the antioxidant activity of apple cubes [104]. A recent study highlighted the finding that the application of VI, using a solution of Ca chloride and pectin methyl-esterase (PME), had a positive effect on firmness, weight loss, soluble solids content, and vitamin C levels of jujube fruits [105], while other authors used VI with solutions of Ca lactate and PME, both alone and in combination, to improve the product’s quality attributes, including texture profile and color, and to prolong the shelf-life of fresh-cut papaya cubes [106]. 

### 2.3. Edible Active Packaging, Based on Natural Compounds

Edible active packaging consists of edible polymers with the incorporation of natural antioxidants. Edible coatings have proven to be an effective primary packaging material in delaying the ripening process, preserving the nutritional properties and preventing quality loss by decreasing several natural processes, including gaseous exchange and the respiration and transpiration rate [107]. Recently, it has been observed that the efficiency of edible coatings can be significantly improved by incorporating active natural components with antioxidant and/or antimicrobial properties. These packaging products are called “active” and are designed to interact with food by releasing components with biological properties. The integration of active compounds into biopolymer matrices enhances the oxidation stability of the food product and inhibits the growth of food-borne pathogens, providing additional safety features for the food products, even in the absence of cold storage [108].

An alginate and chitosan edible coating enriched with an extract from olive leaves has demonstrated the ability to increase the shelf-life of sweet cherries. Moreover, compared with the uncoated samples, a delay in the ripening process and the significant retention of phenolic and antioxidant components were also observed [109]. Robles-Sánchez et al. [110] demonstrated that an alginate coating, when incorporated with citric and acetic acid, allowed fresh-cut mangoes to retain several quality attributes, such as antioxidant activity, phenolic content, and color. However, the flavonoids and 𝛽-carotene could not be retained [110]. Moreover, an edible coating with 1% chitosan and 5% ascorbic acid inhibited browning, retained flesh firmness, delayed microbial growth, and maintained phenolic compounds throughout the storage period in fresh-cut apples [111]. Liu et al. [112] verified the effectiveness of gallic acid-grafted chitosan film as a novel active packaging material for the preservation of mushrooms (*Agaricus bisporus).* Compared to commercial polyethylene film, mushrooms packed with the active film showed significantly lower respiration rate, browning degree, malondialdehyde content, electrolyte leakage rate, superoxide anion production rate, and hydrogen peroxide content, with a potential increase in the antioxidant status of mushrooms, which, in turn, maintained the postharvest quality of the products. Carvalho et al. [113] showed that a 2% chitosan-based coating, enriched with 500 mg L^−1^ antimicrobial *trans*-cinnamaldehyde, preserved the quality of fresh-cut cantaloupe melons during their storage at 4 °C by preserving the content of total vitamin C and carotenoids, lightness, and firmness. Moreover, it was reported that carrots coated with turmeric and casein extended their shelf life by about 7 days, preserving the carotenoid content, the texture, and the antibacterial properties [114]. 

Imeneo et al. [115] demonstrated that a pectin-based coating, enriched with an extract of lemon by-product, ensured that fresh-cut carrots maintained stable structural integrity for 14 days of storage at 4 °C, due to a reduction in enzymatic bacterial activity. This kind of treatment also resulted in higher levels of carotenoids, phenolic compounds, and antioxidant activity. 

Lemon essential oil, incorporated into a chitosan edible coating, has been reported to reduce the respiration rate of strawberries and improve the antifungal activity of chitosan against *Botrytis cinerea* [116]. Ghafoor et al. [117] investigated the effect of chitosan edible coatings with natural functional ingredients obtained from orange peel (OPE) and olive cake (OCE) on the quality attributes of fresh Barhi date fruit. When chitosan was mixed with OPE or OCE, a significant increase in phenolic content and in radical scavenging activity was observed with respect to the uncoated samples. Moreover, chitosan-coating significantly preserved the product’s textural properties, particularly hardness, and inhibited mold growth, without any non-significant changes in the consumer acceptability of the fruit throughout the storage time.

Vieira et al. [118] placed active film pads made from chitosan, enriched by the ethanolic extracts of green tea and rosemary as natural antifungal agents, on the bottom of commercial trays for the preservation of raspberry fruits. This sustainable packaging successfully decreased raspberry fruit fungal incidence, preserved the overall quality during storage, and extended the shelf-life of fruits by up to 14 days.

### 2.4. Strategies of Biocontrol

The study of microbiota represents one of the hottest topics in agri-food research, and this trend is confirmed by the great interest in the microbiota associated with fruit and vegetables [119]. These complex microbial ecosystems encompass the diversity of naturally associated bacteria, yeasts, and filamentous fungi (molds) [120,121]. At harvest, this microbial diversity can play an extremely diversified role, modulating the quality and safety of the postharvest products [122,123]. This microbiota can be associated with undesired microorganism pathogens, the producers of toxins, bacteria-carrying antibiotic resistance genes, and/or with potential spoilage activity with respect to the food matrices of interest [124,125]. At the same time, these undesired microorganisms can arise from postharvest processes, operators, and environments [125]. These spoilage microbes can be controlled using different strategies that are often combined in the framework of hurdle technology applications [125]. Among the other approaches, biocontrol represents the key solution in the field of biological treatments [123,126]. Exploiting selected microbes as control agents, biocontrol is considered one of the more sustainable postharvest approaches to increasing the shelf-life of fruit and vegetables [126,127]. Bio-protection relies on the application of selected microbes that can limit the development of undesired microorganisms. The idea of having the microbiome of the fruit or vegetable as a target offers a privileged perspective in understanding the variables involved in biocontrol solutions, including the currently emerging interest in products that have been developed for preharvest applications but that also demonstrate interest in terms of postharvest biological control [123]. The presence of products on the market (Table 2) helps us to define the diversity of microorganisms that are actually valued in this sector.

All the solutions currently on the market encompass the application of yeast and bacteria as control agents (Table 2). The successful application of yeast in postharvest biocontrol is related to their general predisposition toward physically colonizing the surfaces (also in response to the released exopolysaccharides), efficiently competing for the nutrients to overcome frequently used pesticides, release lytic enzymes, and induce the host resistance [128,129] (e.g., lytic enzymes, Figure 4).

Alongside a global aptitude for concretizing antagonism, there is a diversity of specific killer toxins that can be bio-produced by these eukaryotic microorganisms and that have specific cellular targets in the killer-sensitive microbial cell (e.g., membrane, cell wall, RNA, and replication) [128]. This dual aptitude for creating antagonism justifies the amount of interest in yeasts for postharvest control actions on a wide range of fresh fruit and vegetables (e.g., lemon, see Figure 5).

Even among the prokaryotic microorganisms, steric competition and the antagonism in terms of nutritional factors, as well as the release of lytic enzymes, biofilm formation, and resistance induction (in the host plant), represent the tools of the biological arsenal against postharvest pathogens [130]. To these aspects, we must add the producers of antifungal molecules (e.g., peptides and volatile organic compounds) and the siderophores that chelate iron [130]. The genera *Bacillus* and *Pseudomonas* are considered some of the most effective antagonists for postharvest control [131,132]. *Bacillus* is commonly recognized as a biological alternative to standard chemical fungicides/bactericides in agriculture [132]. *Pseudomonas* has a wide diffusion in the field, attesting to the resilience of this genus in the agricultural environment [131]. Among the different phenotypes of interest in the selected strains, the ability to survive harsh environmental conditions, fast growth (also with low nutrient availability), the capability to modulate the host response, and the production of lytic enzymes/antimicrobial compounds (including the endospore formation for *Bacillus*) have been widely explored [131,132,133]. 

In the field of prokaryotes, the heterogeneous group of lactic acid bacteria (LAB) represents a promising reservoir of potential bio-based solutions for sustainable agriculture, also in terms of postharvest applications [134,135]. LAB generally have a long history of safe use in food fermentations and, also due to this characteristic, several LAB species are recognized as having facilitated paths in defining their safe use in food, according to the different legislative environments [136,137]. The biological mechanisms exploited in the control of undesired microbes on fruits and vegetables after the harvest are multivariate and belong to classes that are widely studied in other food applications [135]. Among the others, this molecular arsenal includes organic acids, bacteriocins, and antimicrobial peptides [138].

All these biomolecules have been exploited to improve the shelf-life of fresh plant products during storage conditions [139,140,141,142], both through the inoculation of cells and/or of the cell-free supernatants [138,143]. There were several matrices, including leafy green, mixed salads, lettuce, potato, mushroom, tomato, melon, cabbage, apple, table grape, lotus root, litchi, strawberry, kiwifruit, and banana, on which the LAB have been successfully applied for this type of purpose [139,142]. The target microorganisms used in the studies are mainly pathogenic bacteria (e.g., *Listeria monocytogenes, Salmonella enterica,* and *Staphylococcus aureus*), but fungal spoilage bacteria (e.g., *Botrytis cinerea, Penicillium expansum,* and *Aspergillus flavus*) have also been employed [142,143,144,145]. It is possible to detect emerging trends regarding the treatment methods used by the selected LABs and/or with molecules having antimicrobial action. One process of particular interest is the use of fermented products, obtained by inoculating the strain of interest [141,145,146,147], or the use of edible coatings [139,148,149]. This is a rapidly evolving field in which there are numerous open problems: from the screening of new strains suitable for maximizing efficacy against a broad spectrum of undesirable microorganisms, to the strategy of applying bacteria to the product, without altering the product’s sensory quality. In general, the application of supernatants and solutions based on the incorporation of the bacterium in a coating (in some cases, this is also of microbial origin) represent interesting solutions as they are able to improve the degree of standardization for these bio-based strategies [139,150,151]. Future perspectives in the field of biocontrol in the postharvest of fresh plant products show that it is crucial to highlight the need for a clear shared global regulatory environment (e.g., [138]) to speed up innovative actions and the assessment of biocontrol, in combination with other physical and chemical solutions (within the framework of hurdle technology) (e.g., [152]).

## 3. Innovative Non-Destructive Techniques for the Quality Monitoring of Fruit and Vegetables

### 3.1. Image Analysis through a Computer Vision System

Computer vision systems (CVSs) are part of an innovative, contactless, and non-destructive technology, based on traditional imaging in the visible range of the electromagnetic spectrum, which is widely used for the in-line grading of fruit and vegetables [153,154,155,156,157,158,159]. CVSs include novel technologies to automatically extract from an image the relevant visual information related to the visual quality of the product: they are used to classify and grade, to assess the quality, to detect defects, and to estimate the internal properties. Proper image analysis algorithms and regression or classification models can perform these tasks. The principal benefits of objective, consistent, and pervasive food control along the entire supply chain, from the producers to the final consumers, that are provided by this technology are a reduction in loss and waste, as well as increased consumer satisfaction. 

A CVS typically acquires images using a setup composed of the combination of a digital camera, an illumination system, and a personal computer that extracts classification features and builds appropriate models, using statistical methods or machine learning approaches (Figure 6).

CVSs have been developed to evaluate the quality and marketability of several fresh commodities since color has proven to be crucial for market acceptance. CVS technology is able to estimate the color properties at the pixel level and to provide a more objective and consistent evaluation with respect to standard colorimeters, which have limited applicability at an industrial level and for in-line monitoring [160].

Recently, an innovative CVS has been developed to evaluate the quality level of rocket leaves and to non-destructively discriminate the cultivation approach using color information extracted from digital images. The proposed CVS permitted the authors to achieve an accuracy of about 95% in quality-level assessment and of about 65–70% in the discrimination of the cultivation approach by the use of a random forest model for the automatic selection of the relevant features for classification [161].

The use of CVS also represents an effective tool to monitor fresh-cut fruit quality along the entire supply chain, reducing food waste and ensuring freshness at the market level. In this context, several interesting applications have been reported in the literature on fresh-cut products: artichokes [162], nectarines [163], iceberg lettuce [153], radicchio [164], apples [165], and potatoes [166]. As regards packaged products, the identification of regions of images affected by the presence of shadows or highlights produced by the interaction of light with a plastic bag is a critical problem that needs to be solved for quality-level assessment to be achieved through the packaging material. A robust and powerful segmentation approach that selects the regions where colors can be measured properly is mandatory to achieve performances similar to the results obtained on unpackaged samples [167,168].

Besides their external appearance, CVS can be used for the evaluation of the inner quality of horticultural products. During postharvest storage, ripening or senescence processes can cause alterations in nutritional quality, leading to changes in the product’s visual attributes, including color and/or texture [169]. In this regard, the yellowing of green leafy vegetables is strictly related to chlorophyll loss, while the browning processes that occur on the surfaces of fresh-cut products are caused by polyphenol oxidase and peroxidase activity on phenolic compounds. Moreover, the total soluble solids and pH values are statistically affected by the various levels of the ripening stages: when the particular fruit color is enhanced, the total soluble solids content increases, and the fruit acidity decreases.

Pace et al. [170], studying the relationships between antioxidant activity (AA) or total phenols (TP) and the color traits of a local landrace of yellow to purple pigmented carrots, developed CVS regression models that were able to estimate the AA and TP contents (R^2^ = 0.97 and R^2^ = 0.94, respectively). 

Moreover, enforcement of the image-processing method, based on the JPEG images of plums at several maturity stages, highlighted the finding that the indices of an RGB color scale of sample images were correlated to the different chemical traits of plums. A robust correlation between fruit acidity (expressed as the total soluble solids content) and the mean intensity of the green color (R^2^ = 0.99) and the R/G ratio (R^2^ = 0.85) was obtained [155].

Recently, image analysis by CVS has been applied to predict the enzymatic activity of both polyphenol oxidase (PPO) and peroxidase (POD) on banana samples, evaluating the peel browning that occurred during 9 days of storage at 25 °C. The extraction of several color features (such as the average and variance of all RGB color parameters) from the images that were acquired by CVS allowed the obtaining of two equations using a genetic programming model that could predict PPO and POD activity during the browning process of banana peels. However, the correlation coefficients did not show significant differences between the predicted and measured values of the enzymatic activity of PPO and POD (R^2^ = 0.98 and 0.97, respectively) [171].

Sabzi et al. [172] implemented an automatic CVS method able to predict with high accuracy (R^2^ = 0.95) the pH value of oranges from images in the visible wavelengths of the external peel. The average time spent in estimating the pH values with the proposed non-destructive technique was 0.42 s, suitable for in-line industrial applications.

A CVS was recently adopted to discriminate between the ripening stages (half-red or red) of strawberries harvested at three different times. Among the chemical indicators of ripening, titratable acidity resulted in it being statistically correlated to the image data (Pearson correlation coefficient = 1), providing a good indicator for the non-destructive evaluation of the ripening stage of strawberries [173].

Additionally, Palumbo et al. [168] proposed a combination of image analysis and the random forest model to forecast the contents of chlorophyll and ammonia, as objective indicators of senescence, for unpackaged and packaged rocket leaves. An irrelevant performance loss on packaged products (Pearson’s linear correlation coefficient was 0.84 and 0.91 for chlorophyll and ammonia content, respectively) compared to unpackaged ones (0.86 for chlorophyll and 0.92 for ammonia) was reported. Moreover, the three partial least squares regression (PLSR) models, built to predict the visual quality level of fresh-cut rocket leaves, using as predictors the total chlorophyll and the ammonia contents obtained by destructive methods, were employed by CVS on packaged products and by CVS on unpackaged ones, highlighting the high performance in validation in terms of R^2^ (0.70, 0.77 and 0.80 for destructive methods, CVS through packaging, and CVS without packaging, respectively).

The application of Image-processing techniques has been recently evaluated for the estimation of the total soluble solids and pH of strawberries. The RGB, HSV, and HSL color-space channels were used as input variables to develop multiple linear regression (MLR) and support vector machine regression (SVM-R) models. The results demonstrated that an SVM-R model working on the characteristics in the HSV color space performed better than an MLR model for total soluble solids and pH prediction (accuracy of 84.1% and 79.2% for total soluble solids and 78.8% and 72.6% for pH in the training and testing stages, respectively) [174].

Finally, one very interesting application of image analysis was reported by Li et al. [175], who provided innovative and smart technology to predict the shelf-life and the quality of kiwifruit during cold storage, via acquiring and calculating the RGB value extracted from photos taken by a smartphone camera. The results showed that the R to B ratio values (Central R/B) were negatively correlated with titratable acidity, vitamin C content, and firmness and were positively correlated with soluble solids content, total soluble sugars, and total plate counts. The smartphone has become an essential portable device nowadays; the methodology proposed by these authors, based on smartphone image analysis, is simpler and faster than the other reported prediction methods. The obtained rapid evaluation of the postharvest quality of kiwifruit may avoid losses and waste, especially for ordinary consumers or fruit retailers at the end of the supply chain.

### 3.2. E-nose

Odors and flavors, crucial sensory parameters for consumer acceptability, are strictly affected by organic volatile compounds (VOCs), which are the final products of the metabolism of plant-based food matrices [176]. The identification and semi-quantification of volatile metabolites via headspace solid-phase microextraction (HS-SPME) sampling, followed by gas chromatography–mass spectrometry (GC-MS), has become a well-established methodology for the assessment of the quality of fruit and vegetables on a molecular basis [177], but the concrete application of HS-SPME/GC-MS in the food industry is limited [178].

In the last few years, the electronic nose (E-nose) has become one of the most favorable sensing technologies as an alternative to conventional HS-SPME/GC-MS. From a practical point of view, it achieves the differentiation and classification of food matrices with different aroma signatures by evaluating the presence and the content of specific volatile metabolites in the headspace of the samples [177,179].

The E-nose is a sensing device supplied with a set of partial specific and broad-spectrum electronic chemical sensors that mimic human olfactory perception; it provides a digital VOC fingerprint that can be explored using suitable statistical tools. These electronic devices are generally composed of three parts: a sample-handling apparatus, a detector, and a system for data acquisition [48]. The use of the olfactometric methodology for the evaluation of the VOC profiles of horticultural food products offers several important advantages, including the low cost of analysis, ease of use, rapidity, non-destructiveness, no need for preliminary sample-preparation steps, environmentally friendly use, and automatic data handling [180,181]. Among the different sensors, the metal oxide semiconductor (MOS) sensors are most frequently used in E-nose applications, presenting the advantages of fast response, high sensitivity, and low cost [182].

Due to the complexity and size of the data matrices produced by the E-nose, it is necessary to treat the sensory responses with chemometric tools to extract information regarding food quality, safety, and fraud recognition and postharvest handling evaluations, as well as ripening assessments [173,183,184,185,186].

Numerous reports have been published on the application of E-noses in the analysis of different foods of plant origin [183]. In particular, according to some recent studies it can be indicated that an E-nose is a useful methodology for investigating the ripening stage of fruit [173,184,187]. 

In the study by Palumbo et al. [173], an E-nose, along with attenuated total reflection-Fourier transform infrared (ATR-FTIR) spectroscopy and image analysis (IA), were used as fast and non-destructive methodologies to discriminate between two ripening stages (half-red or red) of the strawberry variety “Sabrosa”, harvested at three different times. In particular, the PCA carried out on the E-nose values indicated that there was a potential correlation among the E-nose signals and the fruit-ripening stage, as the E-nose device reacted sensitively and selectively to changes in the aroma profiles of strawberries at different maturity degrees. Moreover, the correlation analysis between E-nose data and the VOCs profiles, previously obtained by HS-SPME/GC-MS, demonstrated that E-nose responses were in line with HS-SPME/GC-MS analysis [186,188].

Aghilinategh et al. [184] explored the combination of the responses of a MOS gas sensor E-nose with suitable pattern recognition methods, including artificial neural networks (ANN), principal component analysis (PCA), and linear discriminant analysis (LDA) to determine the maturation stage of white berry and blackberry. Most consistent with the results, although all the statistical methods used were able to differentiate among the fruit, based on the ripeness grades, the ANN showed the best classification performance, with an accuracy of more than 88%.

Recently, Qiao et al. [189] investigated the fruit quality indexes, including titratable acidity, soluble sugar content, sugar–acid ratio, soluble solids amount, soluble protein level, and flavor profile via an E-nose among naturally and artificially ripe crab apples. The aroma patterns generated by 12 MOS sensors were treated by PCA, LDA, SVM, and random forest (RF) analyses. The data obtained indicated that the RF, with an average recognition accuracy of about 98%, can be considered the best algorithm for distinguishing between naturally or artificially ripe crab apples. On the other hand, the correlation analysis, performed by PLSR, among the E-nose values and the quality parameters allowed the authors to establish good predictive models with regression coefficients (R^2^) that were higher than 0.91 [189].

There are several studies reporting the applicability of the E-nose, combined with chemometric strategies, in classifying fruit and vegetables in line with their geographical origin [183]. 

Li et al. [190] investigated the origin of 303 maca samples collected from more than 100 sites within the main growing area in China, using GC-MS and an MOS-based E-nose to detect the maca samples’ volatile and odor fingerprints, respectively. Correlation and multi-regression analyses showed that all sensors had a significant association with specific maca volatiles. Moreover, using E-nose and a backpropagation (BP) neural network algorithm, a maca odor database was implemented that was able to trace the maca origin with predictability greater than 78%. This study suggested that the E-nose is a fast, reliable, and efficient method by which to predict the geographical provenance of Chinese maca [190].

In an interesting paper, data obtained via an MOS E-nose treated with statistical tools were used to discriminate table grapes, depending on the different agronomic practices (conventional vs. organic farming) and geographical origin [191]. PCA experiments performed on the E-nose responses showed poor clustering of the fruit samples according to the growing site or cultivation methods, while a supervised approach using LDA allowed promising prediction rates of 84% and 85% for the discrimination of the growing handling and geographic provenance, respectively [191].

In line with previous reports, the E-nose has been reported to provide effective and useful information for an overall evaluation of the freshness of fruit and vegetables [183,192].

Cozzolino et al. [186] explored the potentiality of the E-nose as a rapid technique in discriminating samples of the sweet cherry variety, “Ferrovia”, packaged in a high-CO_2_ (16% O_2_ + 20% CO_2_ + 64% N_2_) or air (20% O_2_ + 0.03% CO_2_ + 80% N_2_) environment, for up to 21 days. The projection to latent structures (PLS) methods that were applied to the E-nose data indicated that the fresh sample and the packaged or unpackaged fruit could be classified, based on both the storage conditions and the storage time. Furthermore, a correlation analysis among the E-nose responses and the overall VOCs, detected in a previous study via HS-SPME/GC-MS [187] on the same cherry samples, allowed the researchers to associate samples with specific flavor profiles, using one or more E-nose sensors. Specifically, the S10 sensor results were related to some VOCs that are thought to be possible markers of freshness and could, thus, be used for a rapid assessment of product quality. Finally, since the data of the rapid and the conventional approaches agreed, the E-nose is confirmed to be an appropriate method for the real-time physiological and quality evaluations of postharvest plant-based food.

Ghasemi-Varnamkhasti et al. [193] used an E-nose equipped with 8 MOS sensors to evaluate the freshness of strawberries stored in three different polymer packaging types, including polypropylene (PPP), ethylene vinyl alcohol (EVOH), and polyvinyl chloride (PVC). E-nose data, treated with pattern recognition approaches such as PCA, LDA, and SVM, allowed the researchers to properly classify the unpackaged and packaged samples and explore the effects of polymer packages on strawberry freshness. The response surface method (RSM) was considered for the choice of the optimized sensor array with regard to the contribution of each sensor in the sample classification. Sample headspace profiles were studied on days 1, 8, and 16. The results showed that the PCA explained 84% of the variance of the data, while the LDA categorized all the sensor responses with an accuracy of 86.4%. Moreover, the SVM method could accurately distinguish between the samples by 86.4% and by 50.6% in training and validation, respectively, by using a polynomial basis function (C-SVM) and could distinguish between them by 85.2% and 55.6% in training and validation, respectively, using a radial basis function (Nu-SVM). Finally, among the eight sensors used in the study, four of them were selected as the optimal sensors for adoption in an E-nose system.

Huang et al. [194] reported a rapid and non-destructive method to investigate the changes in freshness quality of postharvest spinach from 1 to 12 days of cold storage, using machine vision and an E-nose, combined with chemometrics. Ten trained panelists classified the spinach freshness during cold storage into four grades. K-nearest neighbors (KNN), SVM, and a backpropagation artificial neural network (BPNN) were used for the prediction of spinach freshness. The results obtained from applying the BPNN model related to machine vision showed the same output as the KNN approach, with a classification accuracy of 85.4% in the prediction of spinach freshness. On the other hand, the BPNN model, based on E-nose data, allowed obtaining a better result with respect to the SVM approach, with classification accuracies of 81.2% and 75.0%, respectively. Besides, the BPNN model, built on multisensory data fusion using machine vision and E-nose data, greatly improved the accuracy of the freshness evaluation of postharvest spinach by reaching a classification accuracy of 93.7%.

### 3.3. Near-Infrared Spectroscopy

Infrared (IR) spectroscopy is a very useful technique to recognize specific functional groups in a molecule (Table 3) and, thus, the chemical composition of a product by relating the vibrational properties of matter to certain internal features. Therefore, every sample has a specific IR spectrum; fruit and vegetable products with a similar spectrum also present similar bioactive compounds and nutritional value [195]. Among the IR methodologies, near-infrared (NIR) spectroscopy, which covers the magnetic spectrum range between 780 and 2500 nm, is a rapid, non-destructive, multi-analytical technique that is widely and effectively used in several sectors, such as the food industry, agriculture, chemicals, pharmaceuticals, textiles, polymers, cosmetics, and medical applications [196].

According to the specific application, three setup modes are possible (Figure 7) to use. First, it is important to note that the penetration of NIR radiation into the tissues decreases with the depth, so it is necessary to select the measurement configuration. For example, thick-skinned fruit can compromise reflectance and interactance ability in detecting internal quality or injuries. On the other hand, transmittance can yield information about the skin and core of a fruit, but the process needs high light intensity that can damage the product (Figure 7).

The outcomes of NIR spectroscopy, combined with multivariate techniques and an image analyzer, can yield information about the solid soluble content (SSC), the firmness, the dry matter (DM), the hardness, and, in some cases, the internal injuries of harvested produce.

Peirs et al. [198] examined the spectral variability of apples in reflectance mode and demonstrated that it depends mainly on the seasons and cultivars, and depends less on orchards. Besides, the data variability seems to be associated with the changes in the produce’s texture properties and in its chemical composition. Actually, the sugar and acid content, as well as the cell size, the number of cells, and the number of intercellular spaces were affected by the different growing conditions. Conversely, Schaare and Fraser [199] compared the three NIR modes in estimating the SSC, density, and flesh color of kiwifruit, achieving a higher accuracy by using the interactance mode. The dry matter (DM), the SSC, and the flesh color of kiwifruit were evaluated by Clark et al. [200], applying an interactance modality in order to predict fruit-storage disorder. The model successfully classified the fruits according to their susceptibility to storage, based on the NIR profile. The incidence of postharvest storage disorders increased for fruit that was less mature, which contained less DM, lower SSC, and greener flesh color than their unaffected counterparts. McGlone and Kawano [201] also analyzed the kiwi, obtaining an excellent predictive model for SSC and DM, but not for firmness.

Several studies have focused their research on the internal quality of citrus, specifically. Liu et al. [202] demonstrated the NIR potential to measure the SSC of intact navel orange fruit. This suggests that it can be used as a valid tool for the determination of other internal quality indices in other thick-skinned fruits, such as tangerine and lemon. The results obtained by Gómez et al. [203] confirm NIR as a non-destructive technique for measuring mandarin quality profiling, especially in predicting the sugar content. Similar results were obtained by Lee et al. [204], who evaluated citrus using transmittance methods to yield information about the fruit flesh; in particular, a PLSR model was found to predict sugar content.

Among the NIR techniques, Fourier transform infrared spectroscopy (FTIR) is involved in several studies as it offers the advantages of high sensitivity, high resolution, and fast data acquisition speed. Conversely, FTIR is an expensive and complex tool. Schulz et al. [205] applied NIR-FT-Raman, ATR-IR, and NIR spectroscopy, in combination with chemometric algorithms for the rapid determination of pungency in black and white ground pepper and green whole pepper berries, demonstrating that vibrational spectroscopy methods can replace the conventional procedures for the effective quality control of peppercorns and pepper extracts and of pepper oil treated in the industry. Furthermore, the spectral data can be used to simultaneously classify the pepper oils according to their different monoterpene and sesquiterpene compositions. Amodio et al. [206] applied an FTIR spectrometer to discriminate among strawberries produced by three different fertility management systems. Data treated by chemometric tools allowed the sensor to obtain good classification models to successfully predict TSS, pH, and TA. FTIR, coupled with attenuated total reflectance (ATR-FTIR) techniques, has been demonstrated to present several advantages, including better signal-to-noise ratio, multiplexing, higher energy, and improved resolution [207]. In Palumbo et al. [173], the ATR-FTIR data have been significantly correlated to titratable acidity as a good indicator of the maturity stage of “Sabrosa” strawberries.

The NIR system is an effective non-invasive technique that can be used to analyze the chemical composition of fresh produce and quality changes during storage, promoting its application directly on the field and in the industrial line and as a valid approach for the traceability and authentication of agricultural produce.

## 4. Conclusions

This review paper provides an overview of the effects of advanced postharvest tools (active packaging, dipping, vacuum impregnation, pulsed electric field, high hydrostatic pressure, and cold plasma) and of biocontrol techniques to preserve the high nutritional value and safety of fresh produce after harvesting. 

Physical treatments (such as microwaving, a pulsed electric field, high hydrostatic pressure, and cold plasma) and a biocontrol strategy proved to be useful for improving the safety of products and consequently extending their shelf-life. Technologies such as dipping, vacuum impregnation, and edible active packaging should be applied when the aim of producers is to preserve the quality and enhance the nutritional value of fresh fruits and vegetables. Moreover, the combination of more techniques, among those reported, might positively affect both the safety and the quality of the products. Therefore, in conclusion, the adoption of these technologies represents an innovation in the fruit and vegetable sector, in order to satisfy the consumer's needs. However, cost analysis is necessary to verify the real applicability.

Regarding the non-destructive techniques reported (image analysis, the E-nose, and near-infrared spectroscopy), the research in this field has made it possible to validate its effectiveness for the non-destructive evaluation of fresh and fresh-cut fruit and vegetables from “the field to the fork”, with the aim of optimizing the process phases and limiting losses. Furthermore, the application of these techniques by the end user could increase the satisfaction of the players in the supply chain, improving the quality of the process. 

To this end, portable tools to be applied on industrial lines or in the field are required. Further research is in progress, investigating new technologies to extend shelf life and enable the continuous monitoring of product quality at all stages of the supply chain.

## Figures and Tables

**Figure 1 foods-11-03925-f001:**
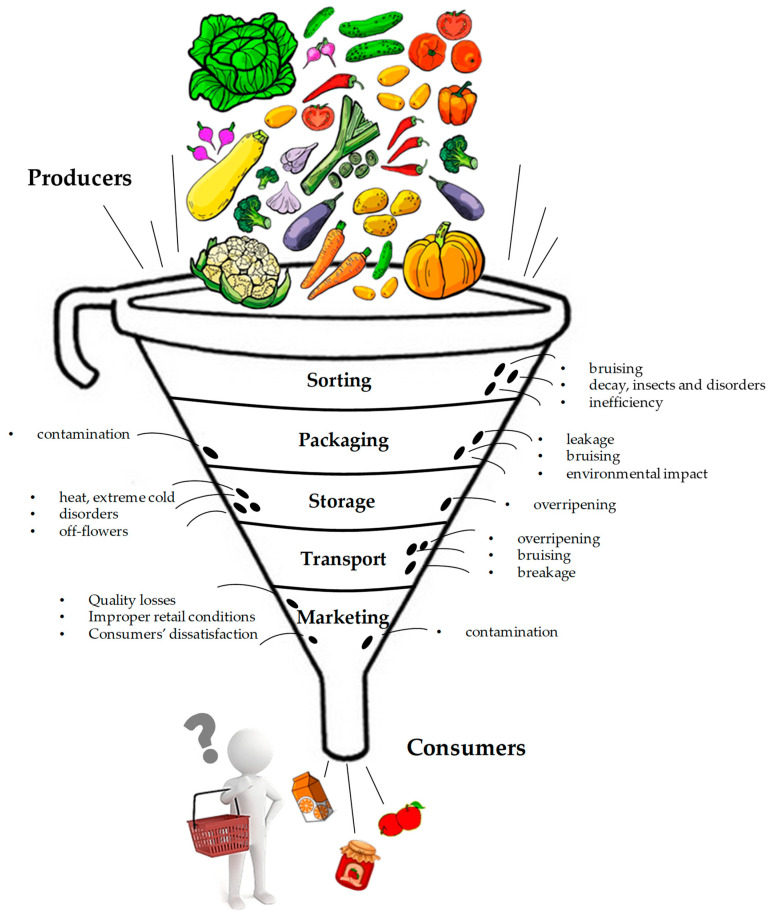
Causes of postharvest losses along the supply chain. (Adapted from [9]).

**Figure 2 foods-11-03925-f002:**
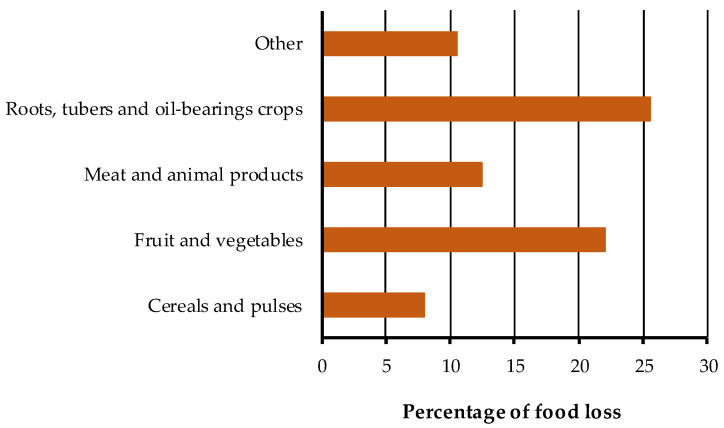
Food losses and waste along the supply chain (percentage for each food group).

**Figure 3 foods-11-03925-f003:**
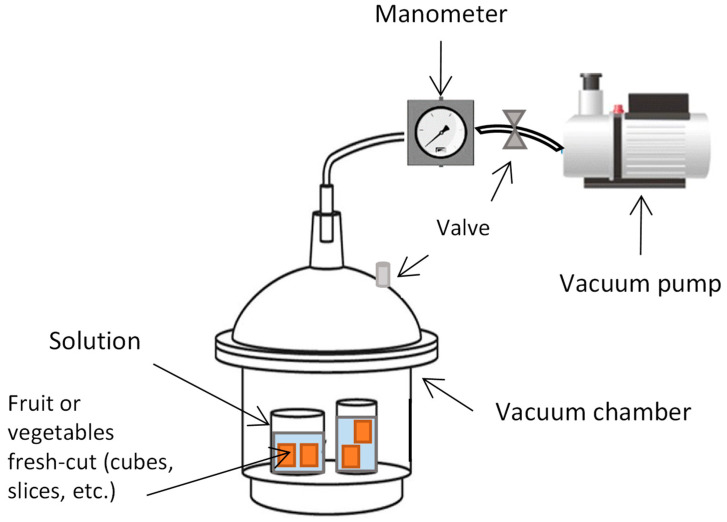
Schematic representation of the vacuum impregnation device, the arrows point at each system element.

**Figure 4 foods-11-03925-f004:**
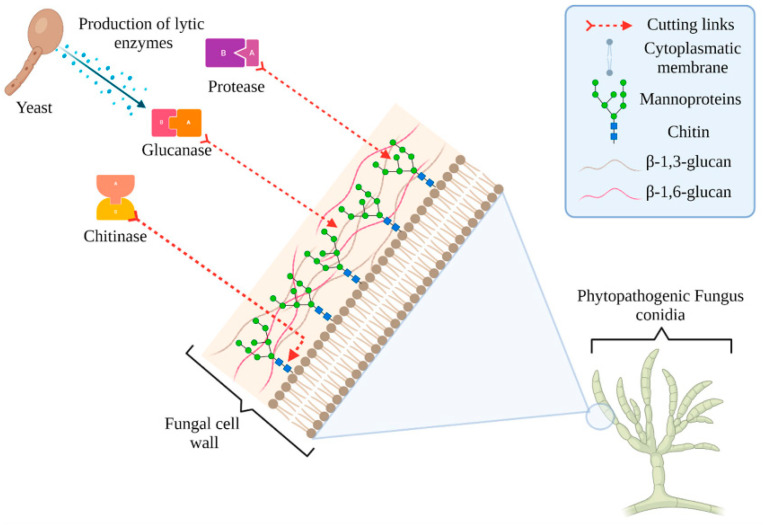
Enzyme production via selected yeasts and their lytic effect on the phytopathogenic fungus cell wall. Reprinted with permission from Hernandez-Montiel et al. [129]. Copyright 2021 MDPI.

**Figure 5 foods-11-03925-f005:**
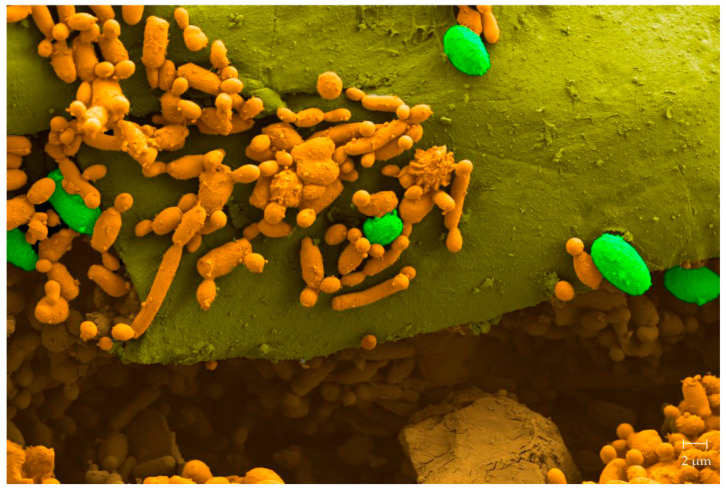
Colored scanning electron microscope (SEM) image of the surface of a lemon wound, inoculated with the killer yeast (orange) and ungerminated spores of citrus fungal postharvest phytopathogen (green). Reprinted with permission from Díaz et al. [128]. Copyright 2020 MDPI.

**Figure 6 foods-11-03925-f006:**
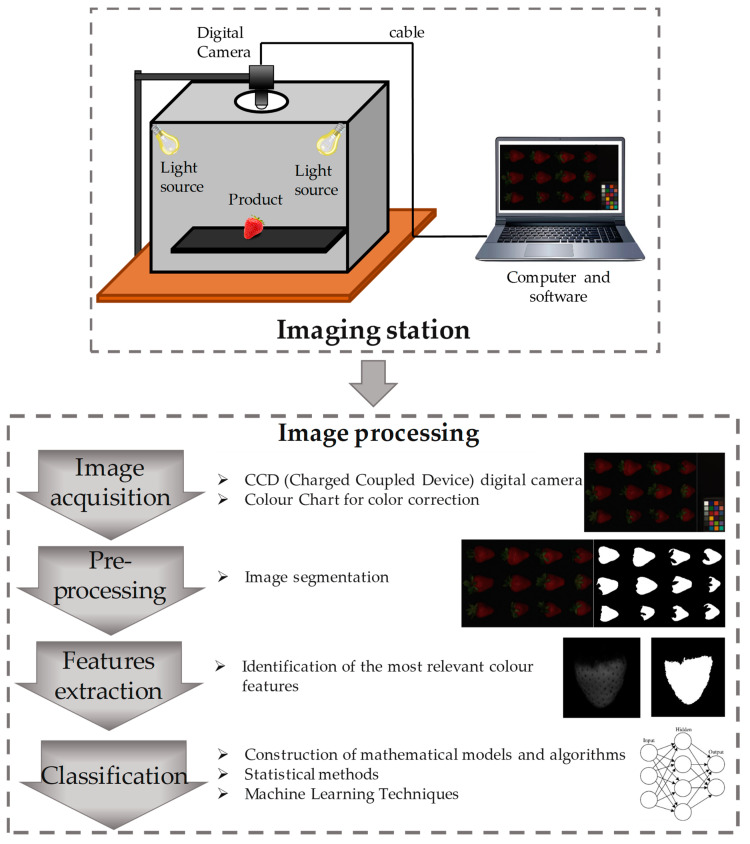
The imaging station and the general workflow of an image-processing system. Four main steps are foreseen in the image processing: Image acquisition, Pre-processing, Features extraction and Classification.

**Figure 7 foods-11-03925-f007:**
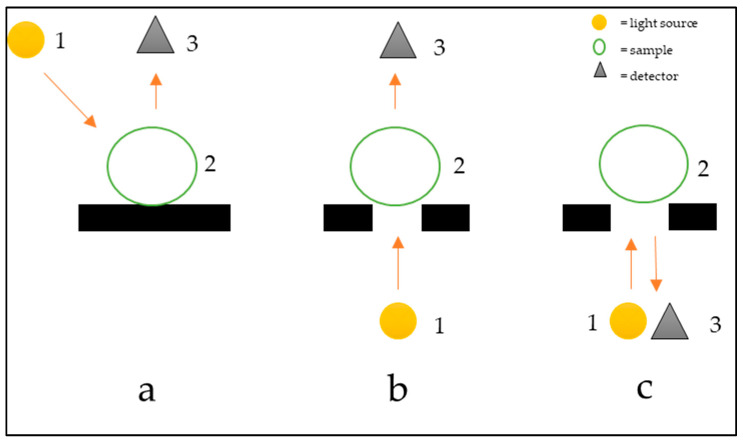
Setup for acquisition in reflectance (**a**), transmittance (**b**), and interactance (**c**). The arrows indicate the direction of light coming out of the lamp (1) hits the sample (2) and then reaches the detector (3).

**Table 1 foods-11-03925-t001:** The main applications of emerging physical postharvest treatments on fresh and minimally processed horticultural products, as reported in research published over the last five years.

Treatment	Experimental Conditions	Food Matrix	Effects	Reference
Microwave	454 W/5 s	Minimally processed bok choy	Microwaving decreases respiration rate, while retarding decay occurrence, and improves cell membrane integrity.	[23]
Microwave	300–100 W/35–10 s	Fresh cut apples	Application of the treatments at the highest intensity after minimal processing led to the best mesophiles and psychrophiles control.	[24]
Pulsed Electric Field	10,000 pulses at 0.5, 1.0 and 1.5 kV cm^−1^	Fresh-cut lotus root	Sugar content was reduced after PEF application in fresh samples, thus lowering the browning index and reducing acrylamide.	[26]
Pulsed Electric Field	5 pulses of 350 kVm^−1^	Carrot	Twelve hours after the treatment, high CO_2_ and volatiles production was observed; after 24 h, the largest total phenolic increase occurred.	[27]
Pulsed Electric Field	0.8, 2 and 3.5 kV cm^−1^ and 5, 12 and 30 pulses	Carrot	Treatment did not affect color, while an increase in the phenolic and carotenoid content and softening was observed.	[28,29]
High hydrostatic pressure	Red cabbage: 150–200 MPa at 35–55 °C, 5–20 min/Radish: 100–200 MPa, at 20–40 °C, 5–10 min	Red cabbage leaves and radish tubers	Variations in the HHP treatments affected the integrity of the tissues and cell turgor.	[30]
High hydrostatic pressure	50–400 MPa for 3–60 min	Fresh-cut papaya fruit	After HHP treatment and storage, the enhancement of carotenoid precursors and carotene content was observed.	[31]
High hydrostatic pressure	100–600 MPa for 2 min	Fresh-cut pumpkin	Color parameters, firmness, electrical conductance, and pectin esterification were positively affected by HHP.	[32]
Mild high hydrostatic pressure	20–80 MPa for 10 min	Mango	HHP reduced the respiration rate, prevented tissue damage, and positively affected the content of bioactive compounds.	[33]
High pressure processing	100–300 MPa for 5–20 min	Minimally processed pineapple	HPP significantly affected firmness, flavonoids, polyphenols, vitamin C content, and colorimetric parameters.	[34]
High pressure treatments	200, 400, 600 MPa for 5 min	Pumpkin	After applying 400 MPa, the pectinmethylesterase enzyme was inactivated. Color parameters decreased and an increase in antioxidant activity was observed.	[35]
High hydrostatic pressures	400–600 MPa; 1–5 min	Blueberries	Treatment caused high tissue damage, thus resulting in the leakage of bioactive cellular components.	[36]
High-pressure treatments	400,600 MPa; 1, 5 min	Zucchini slices	The longest treatment led to more severe cell lysis, browning and dehydration occurrence.	[37]
Dielectric Barrier Discharge (DBD)	5 + 5 min on each side, 10 + 10, and 15 + 15; 30 and 60 min	Fresh cut apple	Plasma treatment caused less browning incidence and enhanced phenols and antioxidant activity after 10 min of treatment.	[38,39]
Cold plasma	40 kV/90 s	Fresh cut cantaloupe	Cold plasma treatment significantly reduced bacteria and mold development during storage. Final product showed higher quality, firmness, and sensory attributes.	[40]
Dielectric barrier discharge cold plasma	Plasma levels of 7% and 14% duty cycle for 5, 10, 20 min.	Strawberry	Plasma treatment of 20 minutes reduced the mesophilic bacteria and yeasts and molds, while not affecting texture and color.	[41]
Atmospheric cold plasma (ACP)	ACP at 60 kV for 10, 15, 30 min	Strawberry	ACP treatment was able to prolong the product shelf-life in treated strawberries. A 15-minute treatment resulted in 2 log bacteria reduction, also enhancing the phenolic content and antioxidant activity. TSS, pH, and moisture were not affected.	[42]
Dielectric barrier discharge cold plasma	60 kV/5 min	Fresh cut pitaya	Total aerobic bacterial count was significantly reduced by treatment, while phenolic content and antioxidant activity increased.	[43]
Atmospheric double barrier discharge plasma	1 or 5 min at 45 and 65 kV	Fresh cut pears	Treatment effectively inhibited the growth of mesophiles and yeast and mold. The 65 kV/1 min treatment slowed respiration rate and maintained organoleptic properties and quality.	[44]
Cold atmospheric dielectric barrier discharge plasma	5, 10, 15 and 20 min	Ready-to-eat rocket leafy salad	A treatment length of 10 min was optimal for the adequate reduction of the microbial load while maintaining color and firmness.	[45]
Atmospheric cold plasma (ACP)	0, 5, 10, 15 and 20 min	Blueberries	ACP treatment inhibited microbial development and decay occurrence. Treatments of 5 and 10 min showed negligible effects on firmness, pH, ORP and anthocyanin concentration, but darkened the color.	[46]
Dielectric barrier discharge gas plasma and arc plasma-activated water (PAW)	20-min DBD treatment and 20 min PAW immersion	Shiitake mushroom	Treatment reduced the total bacterial load after 7 days of storage, also slowing down the overall color modification and positively affecting firmness.	[47]
Plasma-activated water (PAW)	7.0 kHz at 6 kV, 8 kV (PAW-8), 10 kV for 5 min	Fresh cut apple	PAW-8 treatment inhibited bacterial development and the reduced browning of the cut surface without affecting firmness, titratable acidity, radical scavenging activity, and antioxidant content.	[48]
Plasma activated water (PAW)	Activation times of 10, 20, 30, 45 and 60 min, 5 min washing	Fresh cut apple	PAW variably affected enzyme activities, while it did not show an effect on total phenolic content and antioxidant activity. Significant reductions in the aerobic bacteria and yeast and mold loads were observed.	[49]

**Table 2 foods-11-03925-t002:** A non-exhaustive list of biocontrol-based products that are currently marketed for postharvest applications. Adapted with permission from Sellitto et al. [123]. Copyright 2021 MDPI.

Product	Active Ingredient	Country/Company	Fruit/Vegetable	Target
Bio-fungicides recommended for postharvest applications				
Biosave^®^	*Pseudomonas syringae*	Jet Harvest Solutions USA	Pome Fruit, Citrus, Strawberry, Cherry, Potato	*Penicillium*, *Botrytis*, *Mucor*
Nexy^®^	*Candida oleophila*	Lesaffre Belgium	Pome Fruit	*Botrytis*, *Penicillium*
BoniProtect^®^	*Aureobasisium pullulans* (2 strains)	Bio-ferm, Austria	Pome Fruit	*Penicillium*, *Botrytis*, *Monilinia*
BlossomProtect^®^			Grape	
Botector^®^				
Noli	*Metschnikowia fructicola*	Koppert The Netherlands	Table Grape, Pome Fruit, Strawberry, Stone Fruit, Sweet Potato	*Botrytis*, *Penicillium*, *Rhizopus*, *Aspergillus*
Bio-fungicides developed for preharvest applications, also recommended for postharvest				
Serenade^®^ Opti	*Bacillus subtilis*	Bayer	Grape, Berry Fruits, Potato	*Botrytis*, Silver scarf
Amylo-x^®^	*Bacillus amyloliquefaciens*	Biogard, Italy	Grape, Apple, Pear, Kiwifruit	*Botrytis*, *Pseudomonas syringae*
		CBC-Europe, Germany		
Bio-protection agents developed for food processing, also recommended for postharvest				
Gaia™	*Metschnikowia fructicola*	IOC, France	Harvested Grape, Withering Grape, Grape Musts	*Botrytis*, non-*Saccharomyces* spoiling yeasts
Nymphea™	*Torulaspora delbrueckii*	ICV/ Lallemand, France	Harvested Grapes, Grape Musts	*Botrytis*, non-*Saccharomyces* spoiling yeasts

**Table 3 foods-11-03925-t003:** Adsorption peaks of the functional groups. Adapted from [197].

		Adsorption Peaks	
	Functional Group	Wavenumber (cm^−1^)	Wavelength (µm)
O-H	aliphatic and aromatic	3600–3000	2.8–3.3
NH_2_	amine	3600–3100	2.8–3.2
CH	aromatic	3150–3000	3.2–3.3
CH	aliphatic	3000–2850	3.3–3.5
C≡N	nitril	2400–2200	4.2–4.6
C≡C-	alkyne	2260–2100	4.4–4.8
COOR	ester	1750–1700	5.7–5.9
COOH	carboxylic acid	1740–1670	5.7–6.0
C=O	aldehydes and ketones	1740–1660	5.7–6.0
CONH_2_	amide	1720–1640	5.8–6.1
C=C-	alkene	1670–1610	6.0–6.2
Ø-O-R	aromatic	1300–1180	7.7–8.5
R-O-R	aliphatic	1160–1060	8.6–9.4

## Data Availability

Data is contained within the article.

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
