# Peer review of "Emerging Postharvest Technologies to Enhance the Shelf-Life of Fruit and Vegetables: An Overview"

_foods, 2022, doi:10.3390/foods11233925_

Round 1

Reviewer 1 Report

Reviewer comments

Thank you for granting me the opportunity to review this work. In this narrative review, Palumbo et al. reviewed the effect of the application of some emerging technologies on the shelf-life stability of fruits and vegetables. Kindly, find below and attached my comments for your response.

Title:

The authors should please revise the title to capture the effect of the application of the emerging technologies on the shelf-life stability, physicochemical properties and consumer acceptability of fruits and vegetables. That way, it will make the work a “state-of-the art” review. The authors should subsequently have section titles on the effect of the technology application in fruits and vegetables on the shelf-life stability, physicochemical properties and consumer acceptability of fruits and vegetables.

Abstract:

Line 19: The authors’ use of quality in that sentence justifies the need for them to consider tweaking the title as I have suggested above. Also, why would the authors use “cold chain” but not the “farm-to-fork” presentation?

Introduction

In the Introduction, the authors should focus on prioritising their writing on fruits and vegetables and not generalising it to capture all food products. This is because the objective of the work is on fruits and vegetables. The authors should discuss the drivers/factors that cause deterioration.

Line 39: The authors should please indicate the kind of perspective that has changed. A change could be “positive” or “negative”. The authors therefore should be clear in their writing. Also, the quick introduction of the food quality statement is a sharp one.

Line 46: The quality losses is not only attributed to “ripening and senescence processes” but also the high “water activity” which provides a convenient environment for microbial growth. The authors should kindly highlight that. 

Line 47: Instead of horticulture, the authors should please focus on fruits and vegetables.

Line 52: Of the 44% of food losses attributed to poor postharvest practices, the authors should indicate the percentage losses recorded for fruits and vegetables.

Line 79: What are some of the conventional methods for fruits and vegetable preservation?

The Introduction is not strong. Several statements have been put out in the introduction that have got no supporting references to support them. The authors should highlight that. This makes the Introduction a bit speculative. The background picture created is not exhaustive.

The authors should please consider the fact that there are different classes of the vegetables. Examples include cruciferous vegetables and root vegetables. These vegetables differ in terms of their morphology and matrix characteristics. It will therefore be important for the authors to highlight these facts and indicate the various conventional treatment methods/techniques and the emerging technologies employed to extend their storability properties.

This review should be a “state-of-the-art” review and requires that the authors highlight similar reviews that have carried out on the subject matter. From the farm to the fork, it will be expedient for the authors to highlight the various processing methods that can be adopted to extend the shelf-life.

Line 89-91: The aim stated by the authors is at variance with the title of this work. The authors focus is in fruits and vegetables and that must be clearly stated as the aim/objective and not be left in a vague fashion.

Line 97: The authors should kindly consider one technology at a time

Line 104-106: The authors should please provide references

Line 113: The authors should list some of the heating processes that have been employed.

Line 124-127: Kindly, provide a reference

Line 131: replace “to” with “on”

Line 128: The authors fail to highlight the application of the pulsed electric field (PEF) technology on specific vegetables. Also, the authors fail to expand the mechanism of action and which specific vegetables this technology, PEF, has been applied to inactivate microbial activity.

Line 143: No specific vegetables and fruits that had the HPP applied were mentioned.

In the case of the cold plasma technology, even though the authors highlighted that it has been applied to fruits in vegetables including strawberries [34–38], kumquat fruit [39,40] green leafy vegetables [36,41–45], blueberries [31,46–166 48], fresh-cut apples [49–54] pears [55,56],cantaloupe melons [57], mushrooms [58,59], tomatoes [60], kiwifruits [61] and red currants [62], the authors failed to highlight the specific impact the cold plasma technology had on the microbial, textural and sensorial attributes of the products. These should be kindly highlighted.

Line 189: How does the Ca salt inactivate microbial activity?

In section “2.3. Strategies of biocontrol”, the information there is not clear. For example, in the use of the LAB, how can they be practically applied to vegetables and fruits? LAB is a probiotic. Are the authors positing that they will be used to coat the surface of the vegetables and fruits? The authors should also highlight some of the “challenges” associated with the use of such biocontrol methods in fruit and vegetable application. If the process involve direct coating of the fruit and vegetables with the microbes, the authors should also highlight its effect on the sensory profile and acceptability of the fruits and vegetables.

Under this section “3. Innovative non-destructive techniques for quality monitoring of fruit and vegetables”, the authors should be very clear with their presentation. The techniques example the use of “3.1. Image analysis through Computer Vision System” does not by itself affect the shelf-life stability of fruits and vegetables. However, it is part of the quality assurance process that rather helps in sorting out damaged/undersized/oversized fruits and vegetables. With the other methods including E-Nose and Near Infrared spectroscopic methods employed, I see that those technologies were able to profile some of the nutritional, bioactive and aromatic/ volatile compounds. These relates more to the physicochemical properties of the fruits and vegetables and thus will be of great importance if the authors highlight this.

Conclusion

The authors should please revise the conclusion. The authors are not to state the aim/objectives but rather must make the conclusion address the objectives.

Author Response

Response to reviewers

Reviewer 1 comments:

Thank you for granting me the opportunity to review this work. In this narrative review, Palumbo et al. reviewed the effect of the application of some emerging technologies on the shelf-life stability of fruits and vegetables. Kindly, find below and attached my comments for your response.

Title:

The authors should please revise the title to capture the effect of the application of the emerging technologies on the shelf-life stability, physicochemical properties and consumer acceptability of fruits and vegetables. That way, it will make the work a “state-of-the art” review. The authors should subsequently have section titles on the effect of the technology application in fruits and vegetables on the shelf-life stability, physicochemical properties and consumer acceptability of fruits and vegetables.

The title has been modified according the reviewer’s remark.

Abstract:                                                                         

Line 19: The authors’ use of quality in that sentence justifies the need for them to consider tweaking the title as I have suggested above. Also, why would the authors use “cold chain” but not the “farm-to-fork” presentation?

The authors thank the reviewer for the suggestion. The title has been modified, while the expression “from farm to fork” suggested is not appropriate because the present review is only about the postharvest losses, not also losses and waste in the field. Anyway, the sentence has been modified to well explain the general concept.

Introduction

In the Introduction, the authors should focus on prioritising their writing on fruits and vegetables and not generalising it to capture all food products. This is because the objective of the work is on fruits and vegetables. The authors should discuss the drivers/factors that cause deterioration.

Dear reviewer, thanks for the suggestion. The introduction was modified accordingly.

Line 39: The authors should please indicate the kind of perspective that has changed. A change could be “positive” or “negative”. The authors therefore should be clear in their writing. Also, the quick introduction of the food quality statement is a sharp one.

The Authors thank the reviewer. The paragraph has been modified presenting the concept more clearly.

Line 46: The quality losses is not only attributed to “ripening and senescence processes” but also the high “water activity” which provides a convenient environment for microbial growth. The authors should kindly highlight that.

Thank you for the remark. We included a sentence and appropriate references.

Line 47: Instead of horticulture, the authors should please focus on fruits and vegetables.

Done

Line 52: Of the 44% of food losses attributed to poor postharvest practices, the authors should indicate the percentage losses recorded for fruits and vegetables.

The information is added to the text, accordingly.

Line 79: What are some of the conventional methods for fruits and vegetable preservation?

The traditional methodologies have been highlighted in the text.

The Introduction is not strong. Several statements have been put out in the introduction that have got no supporting references to support them. The authors should highlight that. This makes the Introduction a bit speculative. The background picture created is not exhaustive.

Dear reviewer, thanks for the suggestion. The introduction has been modified, accordingly, giving a more exhaustive background.

The authors should please consider the fact that there are different classes of the vegetables. Examples include cruciferous vegetables and root vegetables. These vegetables differ in terms of their morphology and matrix characteristics. It will therefore be important for the authors to highlight these facts and indicate the various conventional treatment methods/techniques and the emerging technologies employed to extend their storability properties.

According to authors’ opinion, an introduction should provide only general information about problems and solutions of the review topic.  Detailed descriptions of the conventional methods/techniques vs the emerging technologies used for the several types of products is discussed into the single paragraphs of the text. Moreover, the aim of the review is to describe specific emerging technologies showing their effect on fruits and vegetables. So, we focused on the technologies and not on products, thus the differentiation in class might be hard for this review.

This review should be a “state-of-the-art” review and requires that the authors highlight similar reviews that have carried out on the subject matter. From the farm to the fork, it will be expedient for the authors to highlight the various processing methods that can be adopted to extend the shelf-life.

Dear reviewer, thanks for the comment. Anyway, the authors have already cited some reviews in the introduction (i.e.  Mahajan et al. 2014; Chauhan et al. 2017) which highlight the common methodologies used to extend the shelf-life. To support this point, the author also added new references on this topic, as suggested by the reviewer (Singh et al. 2014; Ali et al. 2018). To clarify, this review is a collection of the research papers published during the last years about the application of new advanced technologies to extend the shelf-life of fruit and vegetables and, to the best of our knowledge, there are no other papers about this topic in literature, that give a general and detailed background of the technologies discussed in the text.

Line 89-91: The aim stated by the authors is at variance with the title of this work. The authors focus is in fruits and vegetables and that must be clearly stated as the aim/objective and not be left in a vague fashion.

The sentence has been revised, accordingly.

Line 97: The authors should kindly consider one technology at a time

A subparagraph for each technology has been added

Line 104-106: The authors should please provide references

A reference was added according to the reviewer suggestions

Line 113: The authors should list some of the heating processes that have been employed

Some example of heating process has been added in the text

Line 124-127: Kindly, provide a reference

A reference was added according to the reviewer suggestions

Line 131: replace “to” with “on”.

Done

Line 128: The authors fail to highlight the application of the pulsed electric field (PEF) technology on specific vegetables. Also, the authors fail to expand the mechanism of action and which specific vegetables this technology, PEF, has been applied to inactivate microbial activity.

A table containing all the requested information was added

Line 143: No specific vegetables and fruits that had the HPP applied were mentioned.

A table containing all the requested information was added

In the case of the cold plasma technology, even though the authors highlighted that it has been applied to fruits in vegetables including strawberries [34–38], kumquat fruit [39,40] green leafy vegetables [36,41–45], blueberries [31,46–166 48], fresh-cut apples [49–54] pears [55,56],cantaloupe melons [57], mushrooms [58,59], tomatoes [60], kiwifruits [61] and red currants [62], the authors failed to highlight the specific impact the cold plasma technology had on the microbial, textural and sensorial attributes of the products. These should be kindly highlighted.

A table containing all the requested information was added

Line 189: How does the Ca salt inactivate microbial activity?

The sentence has been modified adding the information required by the reviewer.

In section “2.3. Strategies of biocontrol”, the information there is not clear. For example, in the use of the LAB, how can they be practically applied to vegetables and fruits? LAB is a probiotic. Are the authors positing that they will be used to coat the surface of the vegetables and fruits? The authors should also highlight some of the “challenges” associated with the use of such biocontrol methods in fruit and vegetable application. If the process involve direct coating of the fruit and vegetables with the microbes, the authors should also highlight its effect on the sensory profile and acceptability of the fruits and vegetables.

Thank you for the remark. Yes, here, the target of the application is more oriented to the exploitation of LAB that demonstrated a general antimicrobial activity rather than with probiotics, even if, in some strains, the characteristics can coexist. We improved the text following your suggestion, hoping we were able to address properly your indications. 

Under this section “3. Innovative non-destructive techniques for quality monitoring of fruit and vegetables”, the authors should be very clear with their presentation. The techniques example the use of “3.1. Image analysis through Computer Vision System” does not by itself affect the shelf-life stability of fruits and vegetables. However, it is part of the quality assurance process that rather helps in sorting out damaged/undersized/oversized fruits and vegetables.

The authors did never speak about the possibility to directly affect the shelf-life stability of fruits and vegetables by the use of Computer Vision System. So, the concept has been described better from line 434 to 441.

With the other methods including E-Nose and Near Infrared spectroscopic methods employed, I see that those technologies were able to profile some of the nutritional, bioactive and aromatic/ volatile compounds. These relates more to the physicochemical properties of the fruits and vegetables and thus will be of great importance if the authors highlight this.

Concerning the E-Nose, this technique is able to assess the volatile profile of food matrices and discriminate food with different aroma signatures by the detection of the presence and the content of specific volatile components in the headspace of the samples. Consequently, all variations in the physicochemical properties of horticultural products can be related to changes in the volatile patter. This is described from line 563 to line 586 of 3.2 paragraph.

As regard the NIR, a sentence has been added to highlight the concept.

 Conclusion

The authors should please revise the conclusion. The authors are not to state the aim/objectives but rather must make the conclusion address the objectives.

The conclusions have been amended according the reviewer’s suggestion

Reviewer 2 Report

1-The first sentence of the summary is very long and complex.

2-What are the non-contact and non-destructive methodologies could be given in the summary.

3-I like figure 1 a lot :) line 54-55

4-Figure 1 was written in figure 2 and no reference was given.

5-‘non-communicable diseases’ line 38….I didn't understand and it was strange.

6-Line 43…internal factors could be missing…internal (chemical, physical, microbial).

https://dbpedia.org/page/Food_quality#:~:text=This%20includes%20external%20factors%20as,chemical%2C%20physical%2C%20microbial).

7. I would also like to thank you for addressing a very important issue.

8- Line 109 to 112 …this paragraph?

9-No reference is given on Table 2. Line 637

10-Line 515…but their. concrete application…it seems strange.

11-Sentence end point and reference are missing. Line 540

12-If I were to prepare such a compilation, I would first classify the fruits and vegetables within themselves. I would give previous studies and current studies in the light of historical literature in written or tabular form.

Author Response

Response to reviewers

Reviewer 2 comments:

Thank you for granting me the opportunity to review this work. In this narrative review, Palumbo et al. reviewed the effect of the application of some emerging technologies on the shelf-life stability of fruits and vegetables. Kindly, find below and attached my comments for your response.

1 - The first sentence of the summary is very long and complex.

The authors thank the reviewer for the suggestion. The sentence has been simplified.

2 - What are the non-contact and non-destructive methodologies could be given in the summary.

The different types of non-destructive technologies discussed into the review have been added to the summary.

3 - I like figure 1 a lot :) line 54-55.

The authors thank the reviewer. Unfortunately, the authors have substituted this figure with an original one because we did not have the permission for publication.

4 - Figure 1 was written in figure 2 and no reference was given.

Numbers of images have been corrected. The Figure 2 (now Figure 3) is an unpublished image created by the Author.

5 - ‘non-communicable diseases’ line 38….I didn't understand and it was strange.

A non-communicable disease is a disease that is not transmissible directly from one person to another. NCDs include such as cardiovascular disease, cancer, diabetes, Parkinson's disease, autoimmune diseases, Alzheimer's disease and others.

6 - Line 43…internal factors could be missing…internal (chemical, physical, microbial).

 https://dbpedia.org/page/Food_quality#:~:text=This%20includes%20external%20factors%20as,chemical%2C%20physical%2C%20microbial).

The authors thank the reviewer for the suggestion. The sentence has been modified, accordingly.

7- I would also like to thank you for addressing a very important issue.

The authors thank the reviewer.

8- Line 109 to 112 …this paragraph?

The sentence has been modified 

9-No reference is given on Table 2. Line 637

The reference has been added into the text. 

10-Line 515…but their concrete application…it seems strange.

Done

11-Sentence end point and reference are missing. Line 540

 References have been added and text has been amended, accordingly

12-If I were to prepare such a compilation, I would first classify the fruits and vegetables within themselves. I would give previous studies and current studies in the light of historical literature in written or tabular form.

Dear Reviewer, thanks for your suggestion. However, the review was written describing the technologies and their impact on fruit and vegetables’ shelf-life. The different technologies were tested on different products, thus a classification based on fruits and vegetables is complex.

Round 2

Reviewer 1 Report

Reviewer comments:

Thank you for making time to revise the manuscript. Please, find below some very minor comments for your revision.

Line 125-128: The sentence is too long and not too clear. The authors should revise and consider splitting it.

Line 726-728: Provide reference

Line 801: Kindly revise the sentence

Author Response

Thank you for making time to revise the manuscript. Please, find below some very minor comments for your revision.

Line 125-128: The sentence is too long and not too clear. The authors should revise and consider splitting it.

Thank you for the remark. The sentence has been splitted.

Line 726-728: Provide reference

Done

Line 801: Kindly revise the sentence.

Thanks for the suggestion. The sentence has been revised, accordingly.

Reviewer 2 Report

How many is the 1st figure?

Overall, I found the article much more satisfying with the revision. It has been turned into a book section. Thank you.

Author Response

How many is the 1st figure?

The figure 1 has been modified and the first one has been deleted.
